# FedECADO: A Dynamical System Model of Federated Learning

**Aayushya Agarwal** [1]   **Gauri Joshi** [1]   **Larry Pileggi** [1]

## Abstract

Federated learning harnesses the power of distributed optimization to train a unified machine learning model across separate clients. However, heterogeneous data distributions and computational workloads can lead to inconsistent updates and limit model performance. This work tackles these challenges by proposing FedECADO, a new algorithm inspired by a dynamical system representation of the federated learning process. FedECADO addresses non-IID data distribution through an aggregate sensitivity model that reflects the amount of data processed by each client. To tackle heterogeneous computing, we design a multi-rate integration method with adaptive step-size selections that synchronizes active client updates in continuous time. Compared to prominent techniques, including FedProx, FedExp, and Fed-Nova, FedECADO achieves higher classification accuracies in numerous heterogeneous scenarios.

## 1. Introduction

Federated learning collaboratively trains machine learning models across distributed compute nodes, each equipped with a distinct local dataset and computational capabilities (McMahan et al., 2017; Kairouz et al., 2021). Employing techniques from distributed optimization, federated learning samples update from individual clients and calculates an aggregate update to refine the global model. This paradigm yields several advantages, such as improving data privacy by keep data at the edge node, improving model generalizability via collaboration with other nodes, and optimizing efficiency when training large datasets.

Nevertheless, compared to distributed optimization, federated learning encounters unique challenges of non-IID data distributions and varying computation capacities amongst

local clients. Neglecting the impact of heterogeneity can lead to inconsistent models with limited performance (Wang et al., 2020). However, previous research has often studied federated learning in the context where local clients have identical learning rates and an IID data distribution (Kairouz et al., 2021; Wang et al., 2021). As a result, new techniques for aggregating local updates require careful consideration that synchronize heterogeneous client updates for model consistency while being computationally efficient without extensive hyperparameter tuning.

In this work, we design a new consensus algorithm that addresses the challenges of non-IID data distribution and heterogeneous computing in federated learning. We design heuristics inspired by an equivalent circuit model of distributed optimization (Agarwal and Pileggi, 2023) named ECADO (Equivalent Circuit Approach for Distributed Optimization) which represents the trajectory of the global model's state variables by an ordinary differential equation (ODE) and whose steady state coincides with the local optimum. These methods draw inspiration from circuit simulation decomposition methods to address key challenges in federated learning. Our main contributions are:

- **Aggregate Sensitivity Model**: A first-order sensitivity model of each client update provides faster convergence in the consensus step and reflects the fraction of data for non-IID data distributions amongst clients.

- **Multi-rate integration with adaptive step sizes**: Local updates from clients with heterogeneous computational capabilities are synchronized in continuous-time using a multi-rate numerical integration. Using properties of numerical accuracy, we propose an aggregation method with adaptive step sizes for faster convergence.

The key novelty of our work is the introduction of circuit-inspired modeling and simulation principles to federated learning. FedECADO builds on the distribution framework (Agarwal and Pileggi, 2023) and introduces new methodologies (multi-rate integration and aggregate sensitivity model) to address challenges unique to federated settings including intermittent client availability, heterogeneous client computational capabilities, and non-IID data distributions. By reinterpreting these challenges through circuit theory, we are able to use well-established techniques from distributed circuit simulation to handle heterogeneity. This provides a

---

[1]Department of Electrical and Computer Engineering, Carnegie Mellon University, Pittsburgh, USA. Correspondence to: Aayushya Agarwal <aayushya@andrew.cmu.edu>.

*Proceedings of the $42^{nd}$ International Conference on Machine Learning*, Vancouver, Canada. PMLR 267, 2025. Copyright 2025 by the author(s).

new perspective to federated learning for robust and scalable training.

Compared to prominent methods including FedProx, FedExp, FedRS and FedNova, our combined approach demonstrates faster convergence and achieves higher classification accuracies for training deep neural network models in numerous heterogeneous computing scenarios.

## 2. Related Work

Federated learning has received significant attention in addressing challenges related to heterogeneous computation and communication overhead. A comprehensive survey of these methods is available in (Zhu et al., 2021; Kairouz et al., 2021; Wang et al., 2021). We specifically focus on methods that target heterogeneous client computation and non-IID data distributions.

Federated learning methods traditionally rely on discrete iterative algorithms, such as FedAvg (Li et al., 2020) which uses SGD for client training and averages the updated results in each consensus step. While FedAvg offers theoretical guarantees, its performance is often limited in settings with heterogeneous computation and non-IID data distribution (Zhao et al., 2018). To address these limitations, several modifications have been proposed, including adjustments to the SGD update using state information from the central agent (Acar et al., 2021; Pathak and Wainwright, 2020; Karimireddy et al., 2020; Li et al., 2020) and a Newton-like update in FedDANE (Li et al., 2019a). One notable method, FedProx (Li et al., 2020), penalizes client updates deviating far from the central agent step. Other variations of FedProx include the following (Acar et al., 2021; Li et al., 2019b).

Certain federated learning methods have improved convergence rates by introducing new step-size routines (Li et al., 2019b; Malinovsky et al., 2023; Charles and Konečnỳ, 2020) or incorporating momentum into client updates (Das et al., 2022; Xu and Huang, 2022; Khanduri et al., 2021). Adaptive step size selections, as seen in FedYogi, FedADAM, SCAFFOLD (Karimireddy et al., 2020) and FedAdaGrad (Reddi et al., 2020), as well as FedExp (Jhunjhunwala et al., 2023), have also been explored; however, these methods generally do not address heterogeneous computational challenges. A notable method, FedNova (Wang et al., 2020), specifically addresses heterogeneous computation and data scenarios in federated learning by modifying the gradient update to compensate for variations in client local computation.

Our work adopts a new continuous-time formulation of the federated learning process, where the challenges of heterogeneous computation and non-IID data distributions are seen as updates occurring in parallel in continuous time. This enables us to design new methods based on concepts from dynamical system processes. Our continuous-time formulation of optimization is inspired by work on control systems (Behrman, 1998; Attouch and Cominetti, 1996; Polyak and Shcherbakov, 2017; Wilson et al., 2021) and circuit simulation principles (Agarwal et al., 2023; Agarwal and Pileggi, 2023) to design new optimization algorithms. This approach was previously applied to distributed optimization in ECADO (Agarwal and Pileggi, 2023), but was not suitable for federated learning because it assumed full client participation. Our work specifically extends ECADO's ideas to address the distinct challenges posed by federated learning. The key distinctions between our work and ECADO are the introduction of an aggregate sensitivity model to account for non-IID data distributions among clients and a multi-rate integration mechanism to synchronize client updates from heterogeneous client computation.

## 3. Background on Circuit-Inspired Optimization

In federated learning, $n$ distributed edge devices collectively train a global model. Each device, $i$, has a local dataset, $\mathcal{D}_i$, and coordinates with the central server to update the parameters of a global machine learning model, represented by a vector $\mathbf{x}$. Due to communication and privacy constraints, the raw local data is not transferred to the central server; instead only model updates or gradients are shared.

Each client device trains a localized model, where the local objective function $f_i(\mathbf{x})$ is the empirical risk function with respect to the local dataset $\mathcal{D}_i$, defined as

$$f_i(\mathbf{x}) = \sum_{\xi \in \mathcal{D}_i} \ell(\mathbf{x}; \xi), \qquad (1)$$

where $\xi$ is sample index and $\ell(\mathbf{x}; \xi)$ is the sample loss function. The central server seeks to minimize global objective, which is the sum of the local objectives:

$$\min_{\mathbf{x}} f(\mathbf{x}) \text{ where } f(\mathbf{x}) = \sum_{i=1}^{n} f_i(\mathbf{x}). \qquad (2)$$

To tackle the challenges of heterogeneous computing and non-IID data distribution, we design a federated learning algorithm inspired by an equivalent circuit (EC) model of the federated learning process. This EC model builds on the framework introduced in (Agarwal and Pileggi, 2023), which employs a circuit-based approach to distributed optimization. In the EC approach, the solution trajectory of the global objective is analyzed as a continuous-time ordinary differential equation (ODE), referred to as gradient flow:

$$\dot{\mathbf{x}} = -\nabla f(\mathbf{x}) \qquad (3)$$

$$= -\sum_{i=1}^{n} \nabla f_i(\mathbf{x}), \ \ \mathbf{x}(0) = \mathbf{x}_0 \qquad (4)$$

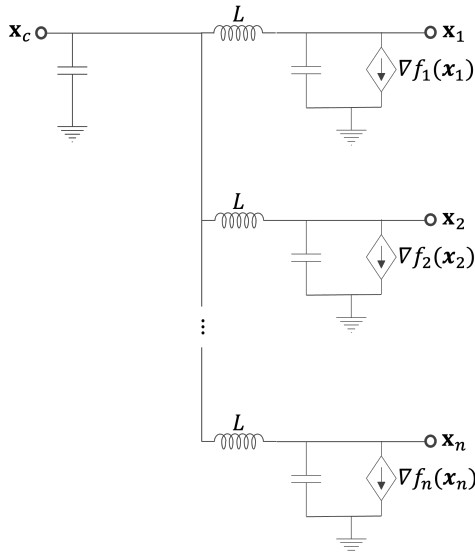

Figure 1: The federated learning process is modeled as an equivalent circuit, where node voltages represent state variables, $\mathbf{x}_i$, and gradients, $\nabla f_i(\mathbf{x}_i)$, are voltage-controlled current sources. Using circuit insights, the gradient flow equations (3),(4) are modified by introducing an inductor (with an inductance of $L$) between the central agent state, $\mathbf{x}_c$, and the state of each sub-problem, $\mathbf{x}_i$. The resulting gradient flow equations (5),(6),(7) are mapped to the equivalent circuit shown.

where $\mathbf{x}_0$ are the initial conditions. At steady-state, the gradient flow (4) reaches a point where $\dot{\mathbf{x}} = 0$, implying $\nabla f(\mathbf{x}) = 0$ as per (3). Thus, the steady-state aligns with a critical point of the objective function (2).

Directly solving the gradient flow equations, (3), to steady-state reveals similar convergence characteristics to those of SGD (Agarwal and Pileggi, 2023). To improve convergence, new insights are derived by translating the ODE into an equivalent circuit (EC) model, motivating physics-based optimization techniques that can effectively address the challenges in heterogeneous computation. In the EC model depicted in Figure 1, the node voltages correspond to the state variables for the central agent and local sub-problems, while the local gradients, $\nabla f_i(\mathbf{x}_i)$, are represented by voltage-controlled current sources. By applying principles from circuit analysis and simulation, we create circuit-inspired algorithms that shape the solution trajectory and select appropriate step sizes. The connection between circuit analysis and the ODEs is provided in Appendix A.

To separate the central agent state, $\mathbf{x}_c$, from the local state of each subproblem, $\mathbf{x}_i$, in the EC model, the gradient flow equations (3),(4) are first modified by introducing an intermediate flow variable (representing an inductor current in Figure 1), $\mathbf{I}_L^i$. The flow variables, $\mathbf{I}_L^i$, interact with the

ODEs of the central agent and local agents according to Kirchhoff's current law (KCL) of the EC as follows:

$$\dot{\mathbf{x}}_c(t) = \sum_{i=1}^{n} \mathbf{I}_L^i(t) \qquad (5)$$

$$\mathbf{I}_L^i(t) + \dot{\mathbf{x}}_i(t) + \nabla f_i(\mathbf{x}_i(t)) = 0 \qquad (6)$$

Inspired by the current-voltage relationship of an inductor, the flow variables, $\mathbf{I}_L^i$, couple the central agent to the local state variables according to:

$$L\dot{\mathbf{I}}_L^{\,i}(t) = \mathbf{x}_c(t) - \mathbf{x}_i(t) \qquad (7)$$

The flow variable, $\mathbf{I}_L^i$, represents the cumulative error between the central agent state, $\mathbf{x}_c$, and local state, $\mathbf{x}_i$, over the simulation window $[0, t]$. This acts as an integral controller for the dynamical system and achieves a second-order effect for faster convergence to steady-state (Agarwal and Pileggi, 2023). At steady-state (i.e., critical point in the optimization function), the system reaches an equilibrium where the flow variables, $\mathbf{I}_L^i$, are stationary, indicating that $\mathbf{x}_c = \mathbf{x}_i$ for all $i$. The settling time for the continuous-time response of the flow variables, $\mathbf{I}_L^i$, is influenced by the hyperparameter $L$ and can be tuned to provide fast convergence as shown in (Agarwal and Pileggi, 2023).

The modified ODEs (5)-(7) describes a set of differential equations with the states of all local subproblems, $\mathbf{x}_i$, implicitly coupled. In federated learning, the differential equations are solved over a set of distributed compute nodes. To decouple the circuit equations (i.e., ODEs), an iterative Gauss-Seidel (G-S) method separates each subproblem from the central agent by treating the intermediate flow variable as constant from the prior iteration. Analyzing the distributed computation as a G-S enables us to study the continuous-time convergence of the full set of ODEs without the effect of discrete updates due to client participation.

The G-S process separately solves each client independently and subsequently communicates updates the coupling variables at each iteration. During the $(k+1)$-th iteration of G-S, a client first simulates its local sub-problem (i.e., gradient flow) as follows:

$$\dot{\mathbf{x}}_i^{k+1}(t) + \nabla f_i(\mathbf{x}_i^{k+1}(t)) + \mathbf{I}_L^{i^k}(t) = 0, \qquad (8)$$

where $\mathbf{I}_L^{i^k}$ is the intermediate flow variable that couples the local client ODE to the central agent. In the G-S iteration, $\mathbf{I}_L^i$ is modeled as a constant from the previous G-S iteration (as indicated by $\mathbf{I}_L^{i^k}$). The differential equation is solved using a numerical integration method over a time window of $[t_0, t_1]$. For example, a Forward Euler integration (equivalent to gradient descent (Agarwal et al., 2023)) solves for the state at each discrete time point:

$$\mathbf{x}_i^{k+1}(t+\Delta t) = \mathbf{x}_i^{k+1}(t) - \Delta t(\nabla f_i(\mathbf{x}_i^{k+1}(t)) + \mathbf{I}_L^{i^k}(t)) \quad (9)$$

where $\Delta t$ is the time step (or learning rate).

During each iteration of G-S, the local subproblem is simulated for a number of time steps, corresponding to a number of iterations denoted as $e_i$. Afterwards, active clients communicate their local states to the central agent. Using the local client updates, the central agent then updates the flow variables, $\mathbf{I}_L^i$, and central agent state, $\mathbf{x}_c$, according to the following ODE:

$$\dot{\mathbf{x}}_c^{k+1}(t) = \sum_{i=1}^n \mathbf{I}_L^{i^{k+1}}(t) \quad (10)$$

$$L\dot{\mathbf{I}}_L^{i^{k+1}}(t) = \mathbf{x}_c^{k+1}(t) - \mathbf{x}_i^{k+1}(t). \quad (11)$$

In this update, the state variables of each sub-problem is represented by a constant value, $\mathbf{x}_i^{k+1}$, which effectively models the sub-problem for a given time period.

The entire circuit then progresses in time towards its natural steady-state. Through this perspective, we design new federated learning algorithms that aims to efficiently simulate the circuit equations (10)-(11) to the critical point of the objective function in the federated learning setting of heterogeneous client computation and non-IID data distributions.

## 4. FedECADO

The EC model depicted in Figure 1 offers a physical analogy to the underlying optimization problem. However, simulating the EC model for federated learning to a steady-state encounters unique challenges due to heterogeneous client computation and non-IID datasets. While prior work (Agarwal and Pileggi, 2023) leveraged insights from the EC model to address homogeneous distributed optimization problems, these approaches are not directly applicable to federated learning and are susceptible to model inconsistencies, as highlighted in (Wang et al., 2020). Specifically, (Agarwal and Pileggi, 2023) simulates the circuit model assuming full client participation and uses a global step learning rate consistent for all clients and the central server. However, heterogeneous computation in federated learning creates asynchronous simulation timescales between clients because each client model is trained using a different learning rate and number of epochs.

We introduce FedECADO, a new algorithm that leverages circuit-based insights from the EC model in Figure 1 to address the challenges posed by heterogeneous client computation and non-IID data distributions in federated learning. Our focus is on deriving the aggregation step for the central server, which processes updates from each active client based on (6). Our approach presents a multirate integration method that synchronizes client updates to account for varying client computational capabilities. Additionally, we propose an aggregate sensitivity approach to model non-IID data distributions.

### 4.1. Aggregate Sensitivity Model

In federated learning, non-IID data distributions can result in inconsistent models if not addressed during the consensus step. To represent the non-IID data distributions, we account for these variations in the following joint optimization with non-IID data distributions:

$$\min_{\mathbf{x}} f(\mathbf{x}) \quad (12)$$

$$f(\mathbf{x}) = \sum_i^n p_i f_i(\mathbf{x}) \quad (13)$$

where $p_i$ scales the contribution of $f_i(\mathbf{x})$ to the overall objective based on the size of local dataset, $\mathcal{D}_i$, relative to the total dataset size, $\mathcal{D}$, defined as:

$$p_i = |\mathcal{D}_i|/|\mathcal{D}|. \quad (14)$$

The gradient flow equations for the federated learning objective with non-IID data distributions (13) is:

$$\dot{\mathbf{x}}(t) = -p_i \nabla f(\mathbf{x}(t)), \quad \mathbf{x}(0) = \mathbf{x}_0. \quad (15)$$

Using the modified gradient flow based on the EC in Figure 1, the set of ODEs describing the circuit are:

$$\dot{\mathbf{x}}_c = \sum_{i \in C} \mathbf{I}_L^i(t) \quad (16)$$

$$\mathbf{I}_L^i(t) + \dot{\mathbf{x}}_i(t) + p_i \nabla f_i(\mathbf{x}_i(t)) = 0 \quad (17)$$

$$L\dot{\mathbf{I}}_L^i(t) = \mathbf{x}_c(t) - \mathbf{x}_i(t), \quad (18)$$

where the local gradients are scaled by the relative dataset size, $p_i$, in (17).

However, the constant-value model of each client assumes that a change in the central agent state, $\mathbf{x}_c$, does not influence $\mathbf{x}_i$ for the time-period. We improve the consensus updates by modeling the first-order effect of each sub-problem due to changes in the state-variable, $\mathbf{x}_c$, using an aggregate sensitivity model of each client's subproblem, denoted by $G_i^{th}$. This reflects the varying data distributions and the relative dataset size, $p_i$, of each client in the consensus step. Incorporating the first-order sensitivity model has been proven to improve the convergence of the G-S process as we can better capture the coupled interactions between clients and adds a proportional controller to the dynamical system to improve the convergence rate of G-S (Agarwal and Pileggi, 2023)(Dartu and Pileggi, 1998).

The aggregate sensitivity model represents a circuit concept known as Thevenin impedance of a client sub-circuit and is defined for the EC in Figure 1 as follows:

$$G_{th}^i = \frac{\partial \mathbf{I}_L^i}{\partial \mathbf{x}_i}. \quad (19)$$

Using (17), we evaluate the sensitivity model, $G_{th}^i$, as

$$G_{th}^i = p_i \frac{\partial}{\partial \mathbf{x}_i} \nabla f_i(\mathbf{x}_i) + \frac{\partial}{\partial \mathbf{x}_i} \dot{\mathbf{x}}_i. \qquad (20)$$

The sensitivity model in (20) includes a partial of a time-derivative term. Assuming a BE step for the ODE, we can numerically evaluate $G_{th}^i$ as:

$$G_{th}^i = \frac{1}{\Delta t} + p_i \nabla^2 f_i(\mathbf{x}_i). \qquad (21)$$

The derivation of $G_{th}^i$ is provided in (Agarwal and Pileggi, 2023). However, calculating the Hessian at each G-S iteration is a bottleneck for computation and communication. To reduce the computation load of evaluating the Hessian, $\nabla^2 f$, for individual datapoints, FedECADO introduces a constant aggregate sensitivity, $\hat{G}_i^{th}$, which is derived by averaging the Hessian across a subset of datapoints. Employing a constant value to approximate the Hessian, $\hat{H} \approx \nabla^2 f_i$, defines a constant sensitivity model for each client:

$$\hat{G}_{th}^i = \frac{1}{\Delta t} + p_i \bar{H}^i \qquad (22)$$

$\hat{G}_{th}^i$ extends the work in (Agarwal and Pileggi, 2023) for the federated learning setting as it captures the relative dataset size, $p_i$, within the first-order sensitivity. Notably, in federated settings with non-IID data, each client's local loss function exhibits a distinct landscape shaped by its data distribution. The Hessian captures the local curvature and is used to estimate the first-order sensitivity of a client's model to changes in the global parameters. This enables the central agent to anticipate how aggregation steps will influence individual client updates.

This sensitivity model can be periodically updated to reassess each client's first-order response, balancing the trade-off between communication and computation. Intuitively, this approach suggests that a client with a larger local dataset will produce a higher $\hat{G}_{th}^i$, thus having a greater influence on the central agent's state updates.

The linear sensitivity model is then incorporated into the G-S process as follows:

$$\dot{\mathbf{x}}_c^{k+1}(t) = \sum_{i=1}^{n} \mathbf{I}_L^{i^{k+1}}(t) \qquad (23)$$

$$L\dot{\mathbf{I}}_L^{i^{k+1}}(t) = \mathbf{x}_c^{k+1} - (\mathbf{I}_L^{i^{k+1}}(t)\hat{G}_i^{th^{-1}} + \mathbf{x}_i^{k+1}(t) - \mathbf{I}_L^{i^k}(t)\hat{G}_i^{th^{-1}}). \qquad (24)$$

The relative values of each client's linear sensitivities drive the central agent states toward certain client updates.

## 4.2. Multi-Rate Integration for Heterogeneous Computation

The aggregate sensitivity model adjusts the ODE equations to accelerate the convergence of the Gauss-Seidel (G-S) process in (23)-(24). To address the challenges posed by heterogeneous client computations, we introduce a technique that synchronizes and simulates the central agent ODEs (23)-(24), for efficient model convergence in federated learning.

During each round of communication, a set of active clients transmit their most recent updates to the central agent. The central agent then numerically solves the ODEs describing the dynamics of the central agent states (23),(24). We apply a Backward Euler (BE) integration step to solve for the states. The BE step is numerically stable and improves the convergence rate of the distributed optimization process (Agarwal and Pileggi, 2023). The BE step solves for the states at a time $t + \Delta t$:

$$\mathbf{x}_c^{k+1}(t + \Delta t) = \mathbf{x}_c^{k+1}(t) - \Delta t \sum_{i=1}^{n} \mathbf{I}_L^{i^{k+1}}(t + \Delta t) \qquad (25)$$

$$\mathbf{I}_L^{i^{k+1}}(t + \Delta t) = \mathbf{I}_L^{i^{k+1}}(t)$$
$$+ \frac{\Delta t}{L}(\mathbf{x}_c^{k+1}(t + \Delta t) - (\mathbf{I}_L^{i^{k+1}}(t + \Delta t)\hat{G}_i^{th^{-1}} \qquad (26)$$
$$+ \mathbf{x}_i^{k+1}(t + \Delta t) - \mathbf{I}_L^{i^k}(t + \Delta t)\hat{G}_i^{th^{-1}})).$$

However, the BE step in (26) assumes a globally synchronous timescale, where all clients are simulated with the same time step, $\Delta t$, for the same number of epochs. This assumption is not applicable to federated learning, where the subset of actively participating clients, $C_a \in C$, exhibit a varying step-size, $\Delta t_i$, and number of epochs, $e_i$.

FedECADO tackles this issue by introducing a multirate integration method grounded in a continuous-time perspective of federated learning. We recognize that in continuous time, each active client simulates its local ODE (6) for a unique time window, $T_i$:

$$T_i = \sum_{k=1}^{e_i} \Delta t_i^k. \qquad (27)$$

where $\Delta t_i^k$ is the learning rate for the client $i$ during an epoch $k$.

This insight builds upon the equivalence between discrete step-sizes and time steps, $\Delta t_i$, resulting in each client essentially simulating its local sub-problem for $e_i$ time steps (i.e., number of epochs). For instance, a client with a local learning rate of $10^{-3}$ and 3 epochs simulates its local ODE for $T_i = 3 \times 10^{-3}$ seconds.

The continuous-time perspective shows that each active client simulates its local ODE on a distinct timescale and

communicates its final state, $\mathbf{x}_i(T_i)$, to the central agent, leading to an asynchronous update. Figure 2 illustrates this issue of asynchronous updates from three active clients. Note, requiring a synchronous timescale is vital for convergence, as all clients must reach steady state simultaneously.

**Remark 1:** Convergence to a critical point for the central agent is achieved when all clients simultaneously reach a steady state.

From a continuous-time point of view, Remark 1 illustrates the importance of maintaining a uniform timescale with each sub-circuit to simultaneously achieve a global steady-state.

Inspired by asynchronous distributed circuit simulation (White and Sangiovanni-Vincentelli, 2012), FedECADO introduces a multi-rate integration scheme designed to address asynchronous local updates in federated learning. This scheme effectively synchronizes the client updates to ensure accuracy and consistency of the central model.

During each communication round, the multi-rate integration scheme begins by collecting the latest updates from all active clients, $\mathbf{x}_i(T_i)$ for all $i \in C_a$, along with each client's simulation runtime, denoted by $T_i$. Communicating the simulation time of each client is essential for synchronizing local updates at the central client server and adds minimal computation and communication costs. Next, FedECADO solves for the central client states on a synchronous timescale at intermediate timepoints. To synchronize the client updates, we employ a linear interpolation and extrapolation operator, $\Gamma(\mathbf{x}_i(t), \tau)$, that estimates client states, $\mathbf{x}_i(t)$, at an intermediate time point, $\tau$, defined as:

$$\Gamma(\mathbf{x}_i(t), \tau) = \frac{\mathbf{x}_i(t_2) - \mathbf{x}_i(t_1)}{t_2 - t_1}(\tau - t_1) + \mathbf{x}_i(t_1), \quad (28)$$

where $\mathbf{x}_i(t_2)$ and $\mathbf{x}_i(t_1)$ represent known state values at time points $t_2$ and $t_1$, respectively.

This constructs a synchronous timescale for the central agent to evaluate its state variables over a time window $\tau \in [t_0, t_0 + \max(T_i)]$, where $t_0$ is the latest time point in the previous communication round and $\max(T_i)$ is the largest simulation time window amongst active clients. The central agent states are now governed by the following ODEs using the operator, $\Gamma(\cdot)$:

$$\dot{\mathbf{x}}_c^{k+1}(\tau) = \sum_{i=1}^{n} \mathbf{I}_L^{i^{k+1}}(\tau) \qquad (29)$$

$$L\dot{\mathbf{I}}_L^{i^{k+1}}(\tau) = \mathbf{x}_c^{k+1}(\tau) - (\mathbf{I}_L^{i^{k+1}}(\tau)G_i^{th^{-1}}$$
$$+ \Gamma(\mathbf{x}_i^{k+1}(t), \tau) - \mathbf{I}_L^{i^k}G_i^{th^{-1}}), \qquad (30)$$

where $\Gamma(\mathbf{x}_i^{k+1}(t), \tau)$ calculates the client states estimated at time $\tau$ using the linear interpolation and extrapolation

operator (28). This operator addresses the challenges posed by asynchronous client updates, which can otherwise lead to model inconsistencies and poor performance. Without the operator, $\Gamma(\mathbf{x}_i^{k+1}(t), \tau)$, the central agent would be forced to incorporate asynchronous client states directly, leading to disparate timescales within the coupled system. This would prevent the central agent and local clients from synchronously reaching a steady state, which has been established by Remark 1 as a necessary condition to converge to a stationary point in the objective function.

FedECADO solves for the central agent states in (29),(30) using a numerically stable BE integration method as follows:

$$\mathbf{x}_c^{k+1}(\tau + \Delta t) = \mathbf{x}_c^{k+1}(\tau) - \Delta t \sum_{i=1}^{n} \mathbf{I}_L^{i^{k+1}}(\tau + \Delta t) \quad (31)$$

$$\mathbf{I}_L^{i^{k+1}}(\tau + \Delta t) = \mathbf{I}_L^{i^{k+1}}(t) + \frac{\Delta t}{L}(\mathbf{x}_c^{k+1}(\tau + \Delta t)$$
$$- (\mathbf{I}_L^{i^{k+1}}(\tau + \Delta t)G_i^{th^{-1}} + \Gamma(\mathbf{x}_i^{k+1}(t), \tau + \Delta t) \qquad (32)$$
$$- \mathbf{I}_L^{i^k}(\tau + \Delta t)G_i^{th^{-1}})).$$

This results in the following set of linear equations that determine the central agent states at the time-point, $\tau$:

$$\begin{bmatrix} 1+\frac{\Delta t \hat{G}_1^{th^{-1}}}{L} & 0 & \dots & -\frac{\Delta t}{L} \\ 0 & 1+\frac{\Delta t \hat{G}_2^{th^{-1}}}{L} & \dots & -\frac{\Delta t}{L} \\ 0 & 0 & \ddots & \frac{-\Delta t}{L} \\ -\Delta t & -\Delta t & \dots & 1 \end{bmatrix} \begin{bmatrix} I_1^{L^{k+1}}(\tau+\Delta t) \\ I_2^{L^{k+1}}(\tau+\Delta t) \\ \vdots \\ \mathbf{x}_c^{k+1}(\tau+\Delta t) \end{bmatrix} =$$
$$\frac{\Delta t}{L} \begin{bmatrix} -\Gamma(\mathbf{x}_1^{k+1}(t),\tau)+\mathbf{I}_L^{1^k}(t)\hat{G}_1^{th^{-1}} \\ -\Gamma(\mathbf{x}_2^{k+1}(t),\tau)+\mathbf{I}_L^{2^k}(t)\hat{G}_2^{th^{-1}} \\ \vdots \\ 0 \end{bmatrix}$$
$$(33)$$

Note, $\Delta t$ represents the learning rate for the central agent and is independent from the client learning rate. To establish the convergence properties of the multi-rate integration using the linear interpolation and extrapolation operator, $\Gamma(\cdot)$, we prove that each central agent step in (31),(32) is a contraction mapping that progressively moves the central agent states toward a stationary point.

**Theorem 4.1.** *The operator $\Gamma(\mathbf{x}, \tau)$, defined in* (28)*, synchronizes local client updates and at each evaluation of the central agent states via the FedECADO consensus step in* (33) *is a contraction mapping towards a stationary point.*

The proof of Theorem 4.1 is provided in Appendix B.

### 4.2.1. SELECTING CENTRAL AGENT STEP-SIZE

During each communication round, we solve the central agent ODEs (29),(30) using a BE integration. The BE integration is a stable numerical method that approximates the central agent state at time points, $\tau \in [t_0, t_0 + \max(T_i)]$

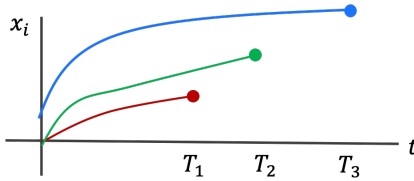

Figure 2: Heterogeneous computation among three clients leads to simulation for different time windows $(T_1, T_2, T_3)$. The final states $(x_1(T_1), x_2(T_2), x_3(T_3))$ are communicated to the central agent, resulting in asynchronous updates.

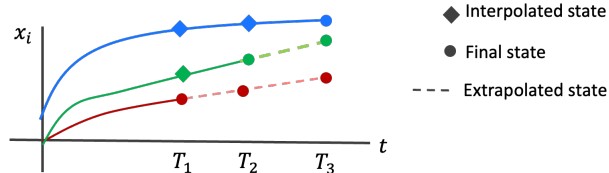

Figure 3: FedECADO proposes a multi-rate integration that evaluates the central agent step at intermediate time points by linearly interpolating and extrapolating client states to the synchronized time point.

over a time-step, $\Delta t$. We propose adaptively selecting $\Delta t$ using numerical accuracy properties of the BE integration.

The accuracy of the BE step for the central agent ODEs ((29) and (30)) can be measured by a local truncation error (LTE) derived in (Pillage, 1998). The LTE for determining the central agent state, $\varepsilon_{BE}^c$, from (29) is estimated as:

$$\varepsilon_{BE}^C = -\frac{-\Delta t}{2}\left[\sum_{i=1}^n \mathbf{I}_L^{i^{k+1}}(\tau) - \sum_{i=1}^n \mathbf{I}_L^{i^{k+1}}(\tau + \Delta t)\right]. \tag{34}$$

The LTE of the BE integration step for evaluating the flow variables from (30), denoted $\varepsilon_{BE}^L$, is estimated as:

$$\varepsilon_{BE_i}^L = -\frac{\Delta t}{2L}[(\mathbf{x}_c^{k+1}(t) - \mathbf{I}_L^{i^{k+1}}(t)\bar{G}_i^{th^{-1}} + \mathbf{x}_i^{k+1}(t) -$$
$$\mathbf{I}_L^{i^k}(t)\bar{G}_i^{th^{-1}}) - (\mathbf{x}_c^{k+1}(t + \Delta t) - \mathbf{I}_L^{i^{k+1}}(t + \Delta t)\bar{G}_i^{th^{-1}} +$$
$$\mathbf{x}_i^{k+1}(t + \Delta t) - \mathbf{I}_L^{i^k}(t + \Delta t)\bar{G}_i^{th^{-1}})] \tag{35}$$

where $\bar{G}_i^{th}$ is the sensitivity model derived in (22).

To accurately capture the ODE trajectory, we adaptively select the time step to guarantee that the accuracy of the BE integration step in (31),(32) remains within a specified tolerance, $\delta$. At each iteration, a backtracking line-search style method (shown in Algorithm 1) selects a step-size, $\Delta t$, to ensure the following accuracy condition is satisfied:

$$\max|\varepsilon_{BE}| \leq \delta, \tag{36}$$

where $\varepsilon_{BE} = [\varepsilon_{BE}^C, \varepsilon_{BE}^L]$.

The adaptive time step selection in Algorithm 1 is initiated by a time step, $\Delta t_0 > 0$, which can be selected as a constant hyperparameter or from the previous communication round. Then a back-tracking line search adjusts $\Delta t$ to ensure that the LTE is bounded by $\delta$. Note that convergence in continuous time guarantees that there exists a $\Delta t > 0$ that satisfies the BE accuracy condition which ensures that the Algorithm 1 is bounded (Agarwal and Pileggi, 2023). Although $\Delta t_0$ is a hyperparameter, it does not affect convergence but can influence the number of iterations in Algorithm 1.

---

**Algorithm 1** Adaptive Time Stepping Method

**Input:** $L > 0, \delta > 0, \Delta t_0 > 0$

1: $\Delta t \leftarrow \Delta t_0$
2: **do while** $\max(|\varepsilon_{BE}|) \leq \delta$
3:      hi
4:      $\Delta t = \frac{\delta}{max(\|\varepsilon_{BE}\|)}\Delta t$
5:      Compute $\mathbf{x}_c^{k+1}(\tau + \Delta t), \mathbf{I}_L^{i^{k+1}}(\tau + \Delta t)$ using (33)
6:      Evaluate $\varepsilon_{BE} = [\varepsilon_{BE}^C, \varepsilon_{BE}^L]$ using (34),(35)
7: Return $\Delta t$

---

## 5. Experiments

We evaluate FedECADO's performance by training multiple models distributed across multiple clients. The FedECADO workflow is shown in Algorithm 2 in Appendix C. We benchmark our approach against established federated learning methods designed for heterogeneous computation, including FedProx (Li et al., 2020), FedNova (Wang et al., 2020), FedExp (Jhunjhunwala et al., 2023), FedDecorr (Shi et al., 2022) and FedRS (Li and Zhan, 2021). Our experiments focus on two key challenges: non-IID data distribution and asynchronous client training. We then demonstrate the scalability of FedECADO on larger models with both non-IID data distributions and asynchronous training in Section D. In these scenarios, FedECADO achieves higher classification accuracy, thus demonstrating its efficacy for real-world federated learning applications.

### 5.1. Non-IID Data Distribution

We evaluate FedECADO's performance by training a VGG model (Simonyan and Zisserman, 2014) on the non-IID CIFAR-10 (Krizhevsky et al., 2009) dataset distributed across 100 clients. To model realistic scenarios, we set an active participation ratio of 0.1, meaning only 10 clients actively participate in each communication round. The data distribution adheres to a non-IID Dirichlet distribution ($\text{Dir}_{16}(0.1)$). The specific dataset size, $|\mathcal{D}_i|$, for each client is predetermined according to the Dirichlet distribution before training and used to precalculate the average sensitivity model proposed in (22). In these experiments,

| Classification Acc. (%) | FedECADO | FedNova | FedProx | FedExp | FedDecorr | FedRS |
|---|---|---|---|---|---|---|
| Mean (Std.) | 57.8 (3.6) | 48.9 (2.9) | 44.3 (3.2) | 45.3 (4.7) | 45.3 (4.7) | 46.1 (4.2) |

Table 1: Classification accuracies for training a VGG-11 model on CIFAR-10 dataset distributed across 100 with Dirichlet data distribution for 100 epochs

| Classification Acc. (%) | FedECADO | FedNova | FedProx | FedExp | FedDecorr | FedRS |
|---|---|---|---|---|---|---|
| Mean (Std.) | 72.2 (3.8) | 67.6 (5.2) | 61.4 (4.8) | 64.6 (3.8) | 60.3 (5.1) | 69.3 (2.8) |

Table 2: Classification accuracies for a VGG-11 model trained on a CIFAR-10 dataset across 100 clients with each client learning rate and epochs set by (37),(38).

| Classification Acc. (%) | FedECADO | FedNova | FedProx | FedExp | FedDecorr | FedRS |
|---|---|---|---|---|---|---|
| Mean (Std.) | 81.3 (4.8) | 69.5 (5.1) | 70.6 (4.1) | 68.6 (10.2) | 61.3 (5.5) | 72.3 (5.8) |

Table 3: Classification accuracies for training ResNet34 model on CIFAR-100 dataset distributed across 100 clients with Dirichlet data distribution and random learning rates for 200 epochs.

the average sensitivity model is not updated during training.

Using each method, we train for 100 epochs, examining the classification accuracy at each step. As illustrated in Figure 4a, FedECADO achieves the highest classification accuracy throughout the training process (with an improvement of 7% compared to FedNova and 13% compared to FedProx). This demonstrates the efficacy of its aggregate sensitivity model in adapting to data heterogeneity.

To test FedECADO's robustness, we repeat the experiment 20 times with random data partitioning sampled by Dirichlet distribution. Table 1 shows the mean and standard deviation (std) of each methods' classification accuracies after 100 epochs. FedECADO exhibits the highest mean accuracy with low variance, demonstrating its effectiveness across diverse data distributions.

### 5.2. Asynchronous Computation

In this experiment, we evaluate the performance of the multi-step integration proposed in Section 4.2. We train the VGG-11 model (Simonyan and Zisserman, 2014) on a CIFAR-10 dataset (Krizhevsky et al., 2009) for 100 epochs across 100 clients with an IID data distribution. However, each client exhibits a different learning rate, $lr_i$, and number of epochs, $e_i$, whose values are sampled by a uniform distribution:

$$lr_i \sim U[10^{-4}, 10^{-3}] \tag{37}$$
$$e_i \sim U[1, 10]. \tag{38}$$

Figure 4b highlights the training loss and classification accuracy for a single random sample of $lr_i$ and $e_i$ using FedECADO, FedNova, and FedProx.

FedECADO's multi-rate integration synchronizes the updates of active clients at each communication round, resulting in faster convergence toward a steady-state and a higher

classification accuracy. Note, due to the IID data distribution, the improvement is solely attributed to the multi-rate integration because $\bar{G}^{th}$ is identical for each client.

FedECADO's improvement is further demonstrated across multiple runs, where the learning rate and number of epochs are randomly selected according to (37),(38). As shown in Table 2, FedECADO achieves a higher mean classification accuracy and the low variance indicates that it performs well across a range of client settings.

### 5.3. Scaling FedECADO for heterogeneous FL

To showcase the effectiveness of our method in heterogeneous settings, we evaluate FedECADO on larger ResNet-34 model trained on CIFAR-100 dataset. In this setup, we study the efficacy of our methodology where the data is distributed according to a non-IID Dirichlet distribution and each client is assigned a random learning rate defined in (37). The result of these experiments are shown in Table 3, where FedECADO further demonstrate its scalability and efficacy in heterogeneous settings as compared to the state-of-the-art optimizers.

**Scaling to Other Datasets and Models**: We demonstrate FedECADO's ability to scale across additional datasets, including Sentiment140 and TinyImageNet, using diverse models such as ResNet-18, ResNet-34, and LSTM in a heterogeneous setting. Appendix D presents the results, comparing FedECADO against FedNova, FedProx, FedExp, FedDecorr, and FedRS.

**Runtime Analysis**: FedECADO achieves a runtime comparable to the baseline methods, as shown in Appendix E.

**Comparison with ECADO**: FedECADO addresses the challenges of heterogeneous federated learning overlooked by ECADO (Agarwal and Pileggi, 2023) and achieves

higher classification accuracies as shown in Appendix F.

## 6. Conclusion

We introduce a new federated learning algorithm, FedECADO, inspired by a dynamical system of the underlying optimization problem, which addresses the challenges of heterogeneous computation and non-IID data distribution. To handle non-IID data distribution, FedECADO constructs an aggregate sensitivity model that is integrated into the central agent update for more accurate model adjustments. To address heterogeneous computation in federated learning, FedECADO employs a linear interpolation and extrapolation algorithm that synchronizes client updates at each communication round. The central model state is then evaluated using a new multi-rate integration, which adaptively selects step-sizes based on numerical accuracy, thus guaranteeing convergence to a critical point. We demonstrate the efficacy of FedECADO through distributed training of multiple DNN models across diverse heterogeneous settings. Compared to prominent federated learning methods, FedECADO consistently achieves higher classification accuracies, underscoring its effectiveness in training distributed DNN models with varying client capabilities and data distributions.

## Impact Statement

This paper presents work whose goal is to advance the field of Machine Learning. There are many potential societal consequences of our work, none which we feel must be specifically highlighted here.

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

## A. Background on Circuit Formulation

Federated learning trains a global model by aggregating updates from distributed clients, each training on local datasets. The update to the global model can be written as:

$$x_{k+1} = x_k - \alpha \sum_{i=1}^{N} \nabla f_i(x_k), \tag{39}$$

where $\alpha$ is the learning rate and $\nabla f_i(x_k)$ is the local gradient from client $i$. As $\alpha \to 0$, this update can be modeled as a continuous-time gradient flow:

$$\dot{x}(t) = - \sum_{i=1}^{N} \nabla f_i(x(t)). \tag{40}$$

This defines an ordinary differential equation (ODE) that assumes each client computes its local gradients using global state, $x_c(t)$. However, in federated settings, each client maintains its own local model state that is periodically communicated to the central agent. To address this, (Agarwal et al., 2023) introduced a continuous-time dynamical model that couples global and client states via auxiliary flow vector, $\mathbf{I}_L(t)$, resulting in the following dynamical system:

$$\frac{d}{dt} x_c(t) + \sum_{i=1}^{|\mathcal{C}|} \mathbf{I}_L^i(t) = 0, \tag{41}$$

$$L_i \dot{\mathbf{I}}_L^i(t) = x_c(t) - x_i(t), \tag{42}$$

$$\dot{x}_i(t) = \mathbf{I}_L^i(t) - \nabla f_i(x_i(t)), \tag{43}$$

where $x_c(t)$ is the global model, $x_i(t)$ is the state of client $i$, and the hyperparameter $L_i$ controls how strongly each client is coupled to the central agent.

The dynamical system is modeled by an ordinary differential equation (ODE) that models the continuous-time evolution of the client and central agent state variables. These state variables converge to a steady-state which coincides with the stationary point of the global objective in (39). By viewing the federated learning setting as an ODE, this reframes the federated learning method as simulating the ODE to its steady-state. However, in general, maintaining accurate and efficient simulation of ODEs is challenging.

To efficiently simulate the ODEs to their steady-state, we adopt methods from circuit simulation, which has demonstrated robust methodologies capable of scaling to billions of transistor devices. We model the dynamical system as an electrical circuit in Figure 1.

In the equivalent circuit model, node voltages represent the model states at each component of the system: the global state, $x_c(t)$, is represented by the voltage at the central server node, while each client's local state, $x_i(t)$, is modeled by the voltage at a corresponding node. These nodes are connected using an electrical device known as an inductor to represent the flow of information between clients and the server.

The dynamics of each client is captured by a capacitor, an electrical component that stores energy and resists sudden changes in voltage. In this context, the capacitor captures the continuous-time evolution of the client's model. The current-voltage behavior of the capacitor is governed by:

$$I_C = C\dot{x}_i(t), \tag{44}$$

where $I_C$ is the total current into the capacitor and $\dot{x}_i$ is the rate of change of the client's state. This reflects how the client integrates incoming signals to update its model parameters.

Each client node is then connected to the central node via an inductor, a component that resists changes in current and introduces momentum-like dynamics into the system. In this equivalent circuit model, the inductor is connected between the central agent node and a client node. Therefore, the voltage across each inductor is given by $x_c(t) - x_i(t)$, representing the difference in the client and global state-variables. The inductor then captures the accumulation of difference between the global and local states over time, and is represented by the following current-voltage relation:

$$x_c(t) - x_i(t) = L_i \dot{\mathbf{I}}_L^i(t), , \tag{45}$$

where $L_i$ is the inductance, and $\mathbf{I}_L^i(t)$ is the current flowing from the server to client $i$. The inductor effectively damps the interaction between client and server, mitigating sharp transitions.

The behavior of the overall system is governed by Kirchhoff's Current Law (KCL), a fundamental principle in circuit theory that states that the total current entering a node must equal the total current leaving it. Applying KCL to each node in the equivalent circuit, we observe that : (1) the current from the central server capacitor equals to the sum from all client branches sum, (2) at each client, the current from the inductor equals the sum of the client capacitor current and gradient-induced currents, $\nabla f_i(x_i)$. This correspondence between circuit behavior and federated optimization dynamics forms the foundation for our simulation-driven approach.

## B. Proof of Theorem 4.1

*Proof.* The convergence proof of FedECADO relies on the following assumptions for each local objective function, $f_{(}\mathbf{x})$.

**Assumption 1.** *(Boundedness)* $f \in C^2$ *and* $\inf_{\mathbf{x} \in \mathbb{R}^n} f(\mathbf{x}) > -R$ *for some* $R > 0$.

**Assumption 2.** *(Coercive)* $f$ *is coercive (i.e.,* $\lim_{\|\mathbf{x}\| \to \infty} f(\mathbf{x}) = +\infty$*)*

**Assumption 3.** *(Lipschitz and bounded gradients): for all* $x, y \in \mathbb{R}^n$, $\|\nabla f(\mathbf{x}) - \nabla f(\mathbf{y})\| \le L\|\mathbf{x} - \mathbf{y}\|$, *and* $\|\nabla f(\mathbf{x})\| \le B$ *for some* $B > 0$.

In order to analyze the convergence of the FedECADO consensus step (33) using the interpolation/extrapolation operator (28), we represent all the state variables of the central agent, including the central agent state $\mathbf{x}_c$ and the flow variables $\mathbf{I}_L^i$, as a vector $X = [\mathbf{I}_L^1, \mathbf{I}_L^2, \cdots, \mathbf{I}_L^n, \mathbf{x}_c,]$. The ODE for the central agent state in FedECADO, as defined by equations (31) and (32), can be generalized as follows:

$$\dot{X}(t) = g(X(t), \mathbf{x}_i(t)), \tag{46}$$

where $g(X)$ is defined as

$$g(X) = \begin{bmatrix} \sum_{i=1}^n \mathbf{I}_L^i(\tau_m) \\ \mathbf{x}_c^{k+1}(\tau_m) - \Gamma(\mathbf{x}_i(t), \tau_m). \end{bmatrix} \tag{47}$$

Here, $\tau_m$ represents the discretized time point indexed by $m$. Furthermore, we generalize the BE integration of the central agent ODE (33) as:

$$\rho(X^{k+1}(\tau_m)) = \Delta t \sigma(g(X^k(\tau_m), \Gamma(\mathbf{x}_1^{k+1}(T_1), \tau_m), \Gamma(\mathbf{x}_2^{k+1}(T_1), \tau_m), \cdots)), \tag{48}$$

where the operator, $\rho(\cdot)$, is defined as

$$\rho = \begin{bmatrix} 1 + \frac{\Delta t G_1^{th^{-1}}}{L} & 0 & \cdots & -\frac{\Delta t}{L} \\ 0 & 1 + \frac{\Delta t G_2^{th^{-1}}}{L} & \cdots & -\frac{\Delta t}{L} \\ 0 & 0 & \ddots & \frac{-\Delta t}{L} \\ -\Delta t & -\Delta t & \cdots & 1 \end{bmatrix}, \tag{49}$$

and $\sigma(\cdot)$, is defined as

$$\sigma = \begin{bmatrix} -\Gamma(\mathbf{x}_1^{k+1}(T_1), \tau_m) + \mathbf{I}_L^{1^k}(\tau_m) G_1^{th^{-1}} \\ -\Gamma(\mathbf{x}_2^{k+1}(T_2), \tau_m) + \mathbf{I}_L^{2^k}(\tau_m) G_2^{th^{-1}} \\ \vdots \\ 0 \end{bmatrix}. \tag{50}$$

Note, the operator, $\rho(\cdot)$, can be inverted to evaluate the central agent states:

$$X^{k+1}(\tau) = \Delta t \rho^{-1} \sigma(g(X^k(\tau_m), \Gamma(\mathbf{x}_1^{k+1}(T_1), \tau_m), \Gamma(\mathbf{x}_2^{k+1}(T_1), \tau_m), \cdots)). \tag{51}$$

The continuous-time ODE of (31),(32) converges to a stationary point characterized by $\mathbf{x}_c = \mathbf{x}_i$ for all $i \in C$ and $\dot{I}_L^i = 0$. The proof of convergence is provided in (Agarwal and Pileggi, 2023). In this analysis, we study the multi-rate discretization of the ODE to ensure the Gauss-Seidel process of solving the coupled system converges toward the steady

state at each iteration. This proof uses the analysis for a multirate waveform relaxation for circuit simulation from (White and Sangiovanni-Vincentelli, 2012). This analysis is based on proving that the central agent step is a contraction mapping towards the steady state (i.e., the stationary point of the global objective function).

The multi-rate integration uses the linear operator, $\Gamma(\cdot)$, to interpolate and extrapolate state variables at intermediate time points. Two important properties of the linear operator are the following:

1. Given two signals, $\mathbf{y}(t)$ and $\mathbf{z}(t)$:

$$\Gamma(\mathbf{y}(t) + \mathbf{z}(t), \tau) = \Gamma(\mathbf{y}(t), \tau) + \Gamma(\mathbf{z}(t), \tau)$$

2. Given a signal $\mathbf{y}(t)$, and a scalar, $\alpha$:

$$\Gamma(\alpha\mathbf{y}(t), \tau) = \alpha\Gamma(\mathbf{y}(t), \tau)$$

To prove convergence of the multi-rate integration step, we employ a continuous-time $\beta > 0$ norm defined as:

$$\|\mathbf{y}\|_\beta = \max_{[0,T]} e^{-\beta t}[\max_i \Gamma(\mathbf{y}_i(t), \tau) \ \ \forall i \in C] \tag{52}$$

Under certain conditions, (51) is a contraction mapping on the $\beta$ norm. To prove this relation, we evaluate the difference between two series, $\{X^k(\tau_m)\}$ and $\{Y^k(\tau_m)\}$, as follows:

$$\{X^{k+1}(\tau_m)\} - \{Y^{k+1}(\tau_m)\} = \Delta t \rho^{-1} \sigma(g(X^k(\tau_m), \Gamma(\mathbf{x}_1^{k+1}(T_1), \tau_m), \Gamma(\mathbf{x}_2^{k+1}(T_1), \tau_m), \cdots)) \\ - \Delta t \rho^{-1} \sigma(g(Y^k(\tau_m), \Gamma(\mathbf{y}_1^{k+1}(T_1), \tau_m), \Gamma(\mathbf{y}_2^{k+1}(T_1), \tau_m), \cdots)). \tag{53}$$

Exploiting the linearity of the operators, $\Gamma(\cdot)$ and $\rho$, leads to the following:

$$\{X^{k+1}(\tau_m)\} - \{Y^{k+1}(\tau_m)\} = \Delta t \rho^{-1} \sigma[g(X^k(\tau_m), \Gamma(\mathbf{x}_1^{k+1}(T_1), \tau_m), \Gamma(\mathbf{x}_2^{k+1}(T_2), \tau_m), \cdots) \\ - g(Y^k(\tau_m), \Gamma(\mathbf{y}_1^{k+1}(T_1), \tau_m), \Gamma(\mathbf{y}_2^{k+1}(T_2), \tau_m), \cdots)]. \tag{54}$$

The BE operator $\rho^{-1}\sigma(\cdot)$ can be expanded into a series of summations, as shown in Appendix B.1, using the following equation:

$$\{X^{k+1}(\tau_m)\} - \{Y^{k+1}(\tau_m)\} = \Delta t \sum_{l=0}^{m} \gamma_l [g(X^k(\tau), \Gamma(\mathbf{x}_1^{k+1}(T_1), \tau_{m-l}), \Gamma(\mathbf{x}_2^{k+1}(T_1), \tau), \cdots) \\ - g(Y^k(\tau), \Gamma(\mathbf{y}_1^{k+1}(T_1), \tau), \Gamma(\mathbf{y}_2^{k+1}(T_1), \tau), \cdots)] \tag{55}$$

where $\gamma_l$ is a scalar that determines the weight of the past state values in the numerical integration method.

To prove that the difference between the series is a contraction mapping, the following two lemmas are useful.

**Lemma B.1.** *Given two sequences, $\{X(\tau)\}$ and $\{Y(\tau)\}$, if $X(T_i) > Y(T_i) \ \forall i$, then $\Gamma(X(T_i), \tau) > \Gamma(Y(T_i), \tau)$. Furthermore, if $X(T_i) = K \ \ \forall i$ where $K$ is a constant value, then $\Gamma(X(T_i), \tau) = K$.*

**Lemma B.2.** *The $\beta$ norm on the following series of $X^K$ is bounded according to the following:*

$$\max_{[0,T]} e^{-\beta \tau} \| \sum_{l=0}^{m} \|\gamma_l \Gamma(X, \tau_{m-l})\| \| \le \frac{M}{1 - e^{-\beta \Delta t}} e^{-\beta \tau} \|\Gamma(X, \tau_m)\|, \tag{56}$$

*where $M$ is equal to $\max_l \|\gamma_l\|$.*

Proof of the two lemmas is provided in Appendixes B.2 and B.3. Using Lemma B.1 and the Lipschitz constant of $\nabla f$, we can bound (55) as follows:

$$\{X^{k+1}(\tau)\} - \{Y^{k+1}(\tau)\} \le |\sum_{l=0}^{m} |\gamma_l| \left[ \sum_{j=1}^{i} \Delta t_i L_{ij} |\Gamma(X_j^{k+1} - Y_j^{k+1}, \tau_{m-l})| + \sum_{j=i+1}^{n} \Delta t_i L_{ij} |\Gamma(X_j^k - Y_j^k, \tau_{m-l})| \right], \tag{57}$$

where $L_{ij}$ is the Lipschitz constant of the $i$th row of $g$ with respect to $X_j$. Using the triangle inequality, we formulate this as follows:

$$\{X^{k+1}(\tau)\} - \{Y^{k+1}(\tau)\} \leq |\sum_{j=1}^{i} \Delta t_i L_{ij} \sum_{l=0}^{m} |\gamma_l||\Gamma(X_j^{k+1} - Y_j^{k+1}, \tau_{m-l})| + \sum_{j=i+1}^{n} \Delta t_i L_{ij} \sum_{l=0}^{m} |\gamma_l||\Gamma(X_j^k - Y_j^k, \tau_{m-l})|. \tag{58}$$

Multiplying both sides by $e^{-\beta t}$ and taking the maximum over the time window, $[0, T]$, results in the following:

$$\max_{[0,T]} e^{-\beta t}\{X^{k+1}(\tau)\} - \{Y^{k+1}(\tau)\} \leq |\sum_{j=1}^{i} \Delta t_i L_{ij} \max_{[0,T]} e^{-\beta t} \sum_{l=0}^{m} |\gamma_l||\Gamma(X_j^{k+1} - Y_j^{k+1}, \tau_{m-l})|$$

$$+ \sum_{j=i+1}^{n} \Delta t_i L_{ij} \max_{[0,T]} e^{-\beta t} \sum_{l=0}^{m} |\gamma_l||\Gamma(X_j^k - Y_j^k, \tau_{m-l})|, \tag{59}$$

where $T = \max(T_i)$. Using Lemma B.2, we conclude the following:

$$\max_{[0,T]} e^{-\beta t}\{X^{k+1}(\tau)\} - \{Y^{k+1}(\tau)\} \leq \left[ \frac{M \Delta t_i}{1 - e^{-\beta \Delta t_i}} \sum_{j=1}^{i} L_{ij} \right] \|X^{k+1} - Y^{k+1}\|_\beta$$

$$+ \left[ \frac{M \Delta t_i}{1 - e^{-\beta \Delta t_i}} \sum_{j=i+1}^{n} L_{ij} \right] \|X^k - Y^k\|_\beta, \tag{60}$$

where $\| \cdot \|_\beta$ is the $\beta$ norm defined in (56).

Assuming all time steps, $\Delta t_i > 0$ are positive and a $\beta > 0$, then there exists a scalar, $\delta > 0$ such that:

$$\delta > \frac{M \Delta t_i}{1 - e^{-\beta \Delta t_i}} \sum_{j=i+1}^{n} L_{ij}. \tag{61}$$

Using the definition of $\delta$, we conclude the following:

$$\max_{[0,T]} e^{-\beta t}\{X^{k+1}(\tau)\} \quad - \quad \{Y^{k+1}(\tau)\} \quad \leq \quad \delta\|X^{k+1} \quad - \quad Y^{k+1}\|_\beta \quad + \quad \delta\|X^k \quad - \quad Y^k\|_\beta. \tag{62}$$

Because the value of $\delta$ holds for all time-steps (indexed by $i$), then:

$$|\{X^{k+1}(\tau)\} \quad - \quad \{Y^{k+1}(\tau)\}|_\beta \quad \leq \quad \delta\|X^{k+1} \quad - \quad Y^{k+1}\|_\beta \quad + \quad \delta\|X^k \quad - \quad Y^k\|_\beta, \tag{63}$$

which can be rewritten as:

$$\|X^{k+1} - Y^{k+1}\|_\beta \leq \frac{\delta}{1 - \delta}\|X^k - Y^k\|_\beta. \tag{64}$$

This proves that for a value of $\delta$ such that $\frac{\delta}{1-\delta} < 1$, the multirate integration scheme, $\rho^{-1}\sigma(\cdot)$, is a contraction mapping whereby the series $\{X^{k+1}(\tau)\}$ and $\{Y^{k+1}(\tau)\}$ converges to a stationary point.

Note, the rate of the contraction mapping, determined by $\frac{\delta}{1-\delta}$, is not affected by the ratio of client step-sizes, $\Delta t_i$. This enables clients to take vastly different step sizes, with accuracy considerations imposed by a Local Truncation Error (LTE). The LTE estimates the error in the approximation of the numerical integration and is a key measure of the accuracy at any iteration.

To prove convergence to a stationary point of the global objective function, we consider the contraction mapping between two series, $\{X^k\}$ and $\{X^*\}$, where $X^*$ is the state at the stationary point (i.e., $g(X^*) = 0$). The difference between the series is diminished according to the contraction mapping:

$$\|X^{k+1} - X^*\|_\beta \leq \frac{\delta}{1 - \delta}\|X^k - X^*\|_\beta. \tag{65}$$

This implies that for a $\Delta t$ ensuring $\frac{\delta}{1-\delta} < 1$, the FedECADO update asymptotically converges to a stationary point, with a convergence rate determined by $\frac{\delta}{1-\delta}$.

$\square$

## B.1. Expanding Backward Euler Operator

Given a general ODE

$$\dot{z}(t) = s(z, t) \tag{66}$$

with an initial known state of $z(0)$, the state of $z$ at a time point, $\tau_m$ is determined by solving the following:

$$z(\tau_m) = z(\tau_{m-1}) + \int_{\tau_{m-1}}^{\tau_m} s(z(\tau), \tau) d\tau. \tag{67}$$

The integral on the right-hand side generally does not have a closed form solution and is approximated using a generalized numerical integration method:

$$z(\tau_m) = z(\tau_{m-1}) + \sum_{l=0}^{n} k_l s(\tau_l, \tau_l), \tag{68}$$

where $k_l$ is a scalar used to weight the contribution of the state at time $\tau_l$. This expression can be further generalized:

$$z(\tau_m) = \sum_{l=0}^{m} \gamma_l s(\tau_l, \tau_l) + z(0), \tag{69}$$

where $\gamma_l$ weights the contribution of past states.

We can apply this form for the Backward-Euler integration step of the ODE (46), which is generally written as follows:

$$X(\tau_m) = X(\tau_{m-1}) + \Delta t_m g(X(\tau_m, x_i(\tau_m)), \tag{70}$$

where the index $m$ represents the iteration of Backward Euler steps taken. An equivalent representation is the following:

$$X(\tau_m) = \sum_{j=0}^{m} \Delta t_{m-j} g(X(\tau_{m-j}, x_i(\tau_{m-j})) + X(0). \tag{71}$$

This expresses the latest state $X(\tau_m)$ as a summation of previous values of $g(\cdot)$.

## B.2. Proof of Lemma B.1

The proof of Lemma B.1 is a direct consequence of the linearity of the operator, $\Gamma$. Consider two sequences, $X(\tau)$ and $Y(\tau)$, which are evaluated at time points $t_1$ and $t_2$, where by:

$$X(t_1) > Y(t_1) \tag{72}$$
$$X(t_2) > Y(t_2). \tag{73}$$

Applying the linear operator, $\Gamma(\cdot, \tau)$ for a time point $\tau \in [t_1, t_2]$, is defined as follows:

$$\Gamma(X, \tau) = \frac{X(t_2) - X(t_1)}{t_2 - t_1}(\tau - t_1) + X(t_1) \tag{74}$$

$$\Gamma(Y, \tau) = \frac{Y(t_2) - Y(t_1)}{t_2 - t_1}(\tau - t_1) + Y(t_1) \tag{75}$$

Because $X > Y$ at time points $t_1$ and $t_2$, we observe the following:

$$\frac{X(t_2) - X(t_1)}{t_2 - t_1}(\tau - t_1) + X(t_1) > \frac{Y(t_2) - Y(t_1)}{t_2 - t_1}(\tau - t_1) + Y(t_1) \tag{76}$$

thereby proving that $\Gamma(X, \tau) > \Gamma(Y, \tau)$. Expanding this proof to multiple evaluated time points $T_i$, we note that if $X(T_i) > Y(T_i) \forall i$, then $\Gamma(X, \tau) > \Gamma(Y, \tau)$.

### B.3. Proof for Lemma B.2

The proof of Lemma B.2 is reconstructed in the following from [(White and Sangiovanni-Vincentelli, 2012)].

From the definition of the $\beta$-norm, we see that:

$$\|\sum_{l=0}^{m} \gamma_l X(\tau_{m-l})\|_\beta = \max_m e^{-\beta \Delta t m} \|\sum_{l=0}^{m} \gamma_l X(\tau_{m-l})\|, \tag{77}$$

which can be upper-bounded using the triangle inequality as:

$$\|\sum_{l=0}^{m} \gamma_l X(\tau_{m-l})\|_\beta \leq \max_m e^{-\beta \Delta t m} \sum_{l=0}^{m} |\gamma_l| \|X(\tau_{m-l})\|. \tag{78}$$

Multiplying $e^{\beta(m-l)\Delta t} e^{-\beta(m-l)\Delta t}$ (equal to 1) into the right-hand side of the equation above leads to

$$\|\sum_{l=0}^{m} \gamma_l X(\tau_{m-l})\|_\beta \leq \max_m e^{-\beta \Delta t m} \sum_{l=0}^{m} |\gamma_l| e^{\beta(m-l)\Delta t} e^{-\beta(m-l)\Delta t} \|X(\tau_{m-l})\|. \tag{79}$$

Because $e^{-\beta(m-l)\Delta t} \|X(\tau_{m-l})\| \leq \|X(\tau_{m-l})\|_\beta$, then

$$\|\sum_{l=0}^{m} \gamma_l X(\tau_{m-l})\|_\beta \leq \max_m e^{-\beta \Delta t m} \sum_{l=0}^{m} |\gamma_l| \|X(\tau_{m-l})\|_\beta. \tag{80}$$

Suppose $|\gamma_l|$ is upper-bounded by $M$, then the inequality becomes

$$\|\sum_{l=0}^{m} \gamma_l X(\tau_{m-l})\|_\beta \leq M \sum_{l=0}^{m} [e^{-\beta \Delta t m}] \|X(\tau_{m-l})\|_\beta \tag{81}$$

Because $e^{-\beta \Delta m} > 0$ and $\sum_{l=0}^{m} e^{-\beta \Delta m} \leq \sum_{l=0}^{\infty} e^{-\beta \Delta m}$, then the inequality is as follows:

$$\|\sum_{l=0}^{m} \gamma_l X(\tau_{m-l})\|_\beta \leq M \sum_{l=0}^{\infty} [e^{-\beta \Delta t m}] \|X(\tau_{m-l})\|_\beta. \tag{82}$$

The infinite series can be directly computed as:

$$\sum_{l=0}^{\infty} e^{-\beta \Delta m} = \frac{e^{-\beta \Delta t}}{1 - e^{-\beta \Delta t}}, \tag{83}$$

where the upper bound is as follows:

$$\|\sum_{l=0}^{m} \gamma_l X(\tau_{m-l})\|_\beta \leq M \frac{e^{-\beta \Delta t}}{1 - e^{-\beta \Delta t}} e^{-\beta \Delta t m}] \|X(\tau_{m-l})\|_\beta, \tag{84}$$

which proves Lemma B.2.

## C. FedECADO Algorithm

The full workflow for the FedECADO algorithm is shown in Algorithm 2. The algorithm begins by initializing the state values in Steps 1–4 and precomputing the constant sensitivity model, $\bar{G}_{th}$, for all clients in Step 5. The hyperparameters for the algorithm are $L > 0$ and the local truncation error tolerance, $\eta > 0$.

For each epoch, FedECADO begins by simulating the set of active clients, $C_a \in C$, in step 10, for a number of epochs, $e_i$. The client ODE is solved by using numerical integration selected by the user (further details are provided in (Agarwal et al., 2023)). In step 11 of Algorithm 2, we use a Forward-Euler integration to simulate the local ODE.

The active clients then communicate their final states, $\mathbf{x}_i^{k+1}(t + T_i)$, and simulation time, $T_i$, to the central agent server, which evaluates its own states at intermediate time points (steps 12-16). First the central agent estimates the active client state values at a time point, $\tau$, using the operator, $\Gamma(\cdot, \tau)$. Then after selecting a time step, $\Delta t$, that satisfies the accuracy conditions in (34),(35), the central agent solves for its states at the proceeding time point, $\tau + \Delta t$, in step 15. The central agent server progresses through time (performing steps 13-14), until it has simulated the maximum client simulation time window determined by $\max(T_i)$.

Note, for a given time-step, $\Delta t$, the central agent LU-factorizes the left hand matrix in Step 16 of Algorithm 2 (33), so that any subsequent central agent steps with the same step size only requires a forward-backward substitution to solve for the central agent states. In practice, once an appropriate step size, $\Delta t$, is selected that satisfies the accuracy conditions in (34)-(35), it is infrequently updated. As a result, re-using the same LU-factor provides a computational advantage across multiple central agent evaluations and improves the overall runtime performance of FedECADO.

---

**Algorithm 2** FedECADO Central Update

---

**Input:** $\nabla f(\cdot), \mathbf{x}(0), \eta > 0, L > 0$

1: $\mathbf{x}_c \leftarrow \mathbf{x}(0)$
2: $\mathbf{x}_i \leftarrow \mathbf{x}(0)$
3: $I_i^L \leftarrow 0$
4: $t \leftarrow 0$
5: Precompute $\bar{G}_i^{th} \ \forall i \in C$
6: **do while** $\|\dot{\mathbf{x}}_c\|^2 > 0$
7: $\quad$ $\mathbf{x}_c^k \leftarrow \mathbf{x}_c^{k+1}$
8: $\quad$ $\mathbf{x}_i^k \leftarrow \mathbf{x}_i^{i^{k+1}}$
9: $\quad$ *Parallel Solve for active client states,* $\mathbf{x}_i^{k+1}(t + T_i) \forall i \in C_a$, *by simulating:*
10: $\quad\quad$ **for** $e_i$ **epochs:**
11: $\quad\quad\quad$ $\mathbf{x}_i^{k+1}(t + \Delta t_i) = \mathbf{x}_i^{k+1}(t) - \Delta t_i \nabla f(\mathbf{x}_i^{k+1}(t)) - \Delta t_i I_i^{L^k}(t)$
12: $\quad\quad$ **for** $\tau \in [t, t + \max(T_i)]$
13: $\quad\quad\quad$ Select $\Delta t$ according to Algorithm 1
14: $\quad\quad\quad$ Evaluate active client states at timepoint $\tau$: $\Gamma(x_i^{k+1}, \tau) \ \forall i \in C_a$
15: $\quad\quad$ *Solve for* $\mathbf{x}_c^{k+1}(\tau + \Delta t), I_i^{L^{k+1}}(\tau + \Delta t)$ *according to* (33)
16: $\quad\quad\quad$ $\tau = \tau + \Delta t$
17: Return $\mathbf{x}_c$

---

## D. Scaling the FedECADO Algorithm

The previous experiments highlight the individual contribution of the proposed aggregate sensitivity model and multi-rate integration on distributed training. In this experiment, we study the impact of both methods to address the challenges of federated learning across multiple datasets and models.

We demonstrate the effectiveness and scalability of FedECADO by training multiple models (ResNet 34, ResNet18 and LSTM) on datasets including CIFAR-100, TinyImageNet and Sentiment140, distributed among 100 clients using a Dirichlet allocation. In this setup, only 10% of clients participate in each training round, and each client is assigned a random learning rate as defined in equation (37). We train a larger ResNet-34 model on a CIFAR-100 dataset distributed across 100 clients with both non-IID data distribution as well as heterogeneous learning rates and numbers of epochs, determined by (37),(38). Tables 3, 4, 5 demonstrates the efficacy of FedECADO and larger and varied datasets and models.

Figure 5 showcases FedECADO's advantage over FedNova and FedProx in training the ResNet-34 model. Our approach achieves a lower training loss, indicating more efficient convergence, as well as higher classification accuracy after 100 epochs (4.6% higher than FedNova and 8.6% higher than FedProx).

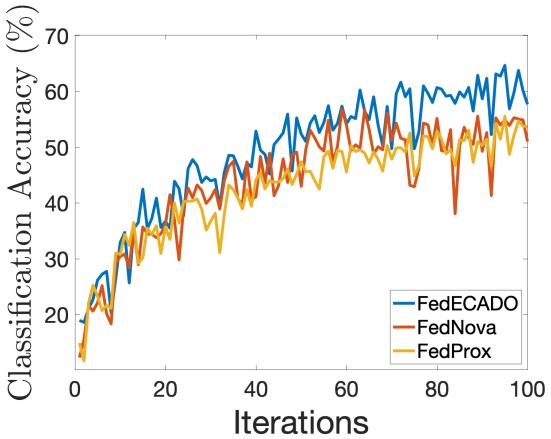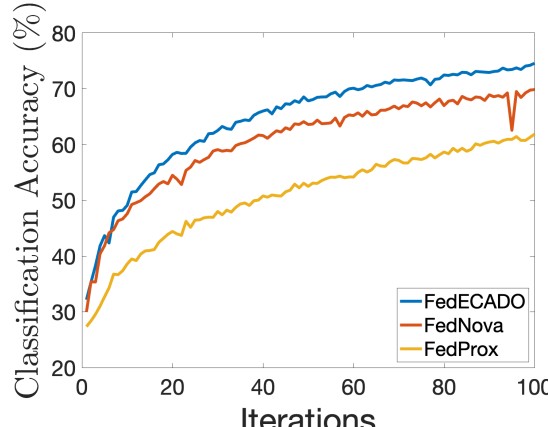

Figure 4: The training loss and classification accuracy for a VGG-11 model trained on a CIFAR-10 dataset across 100 clients with (a) non-IID Dirichlet distribution and identical learning rates, and (b) each client's learning rate and number of epochs is randomly determined by (37),(38) .

| Classification Acc. (%) | FedECADO | FedNova | FedProx | FedExp | FedDecorr | FedRS |
|---|---|---|---|---|---|---|
| | 58.9 | 45.4 | 40.7 | 47.6 | 41.9 | 44.3 |

Table 4: Classification accuracies for training a ResNet-18 on TinyImageNet dataset distributed across 100 clients with Dirichlet data distribution and random learning rates for 60 epochs.

| Testing Errors | FedECADO | FedNova | FedProx | FedExp | FedDecorr | FedRS |
|---|---|---|---|---|---|---|
| | 79.9 | 77.1 | 78.6 | 78.4 | 79.2 | 77.4 |

Table 5: Testing errors for training a LSTM model [R2] on Sentiment140 dataset distributed across 10 clients with Dirichlet data distribution and random learning rates for 10 epochs.

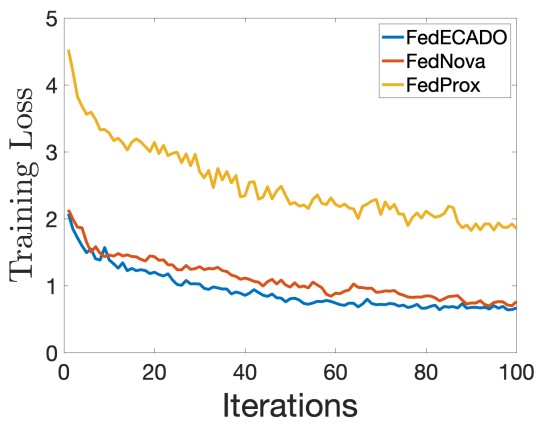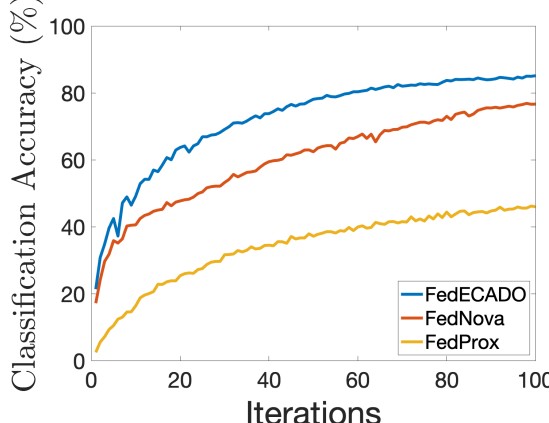

Figure 5: Scaling FedECADO to train ResNet34 model on CIFAR-100 dataset distributed on 100 clients with heterogeneous computation (non-IID data distribution and asynchronous updates.

## E. Runtime Analysis

Despite evaluating the central agent multiple times for multi-rate integration, FedECADO's clock time for centralized updates is comparable to those of FedNova and FedProx (only 1% slower than FedNova and 2.4% slower than FedProx). The complete comparison of runtime analysis is shown in Table 6. The main computational cost of Algorithm 2 occurs on the subset of active clients during each communication round, which is significantly smaller than the total client base. The resulting BE matrix (33) is relatively small, with dimensions of $\mathcal{R}^{a \times a}$, where $a$ is the size of the active client list. Additionally, we pre-compute the LU matrix of in (33) to minimize the computation time for central agent evaluations.

Furthermore, a potential bottleneck for FedECADO's runtime can be attributed to the adaptive server step sizes routine in Algorithm 2. This routine is initiated by the step size from the prior communication round and in practice, does not require multiple iterations to satisfy the LTE conditions in equation (34) and (35).

|  | FedECADO | FedNova | FedProx | FedExp | FedDecorr | FedRS |
|---|---|---|---|---|---|---|
| Normalized Runtime | 1.06 | 1.0 | 1.02 | 1.04 | 1.02 | 1.0 |

Table 6: Normalized per-epoch runtime (normalized to FedProx) of a centralized server step for training VGG model on CIFAR-10 dataset distributed across 100 clients.

## F. Comparison with ECADO

FedECADO uses the distributed optimization method from ECADO (Agarwal and Pileggi, 2023) as a basis to derive insights for the unique challenges of federated learning including heterogenous dataset distributions and varying client computational capabilities (i.e., learning rates). However, ECADO is not equipped to handle the heterogenous challenges of federated learning, and is prone to model drifts, an issue demonstrated in (Wang et al., 2020). Nonetheless, the unique perspective of an equivalent circuit model of the federated learning process allows us to derive intuitive methodologies. For example, the challenge of heterogeneous client learning rates is clearly identified as a challenge of asynchronous communications in the continuous-time representation. From the circuit perspective which views the distributed computation as splitting a large circuit, the immediate solution to this problem is using an interpolation/extrapolation operator to synchronize client computations, as described in Section 4.2.

As a result of these improvements, we observe that FedECADO improves the convergence of ECADO in the federated learning setting. The difference is shown in in Table 7, where we observe a significant improvement in performance when training distributed CIFAR-100 dataset with non-IID data distributions and varying client learning rates.

| Classification Acc. | FedECADO | ECADO |
|---|---|---|
| Mean (std) % | 81.3 (4.8) | 76.5 (2.1) |

Table 7: Classification accuracies for training a ResNet-34 model on CIFAR-100 dataset distributed across 100 clients with Dirichlet data distribution and random learning rates for 200 epochs.

