# OpenReview forum: "FedECADO: A Dynamical System Model of Federated Learning"
_ICML.cc/2025/Conference — ICML 2025 poster_

### Official Review · Reviewer_hJyM · 2025-03-08

**Overall Recommendation:** 3

**Summary:**

This work addresses the inherent challenges of heterogeneous data distributions and computational resource disparities in FL by introducing FedECADO, a novel algorithm inspired by a continuous-time ODE theoretical framework for understanding federated optimization process. Extensive empirical studies have demonstrated the effectiveness of the proposed method.

## update after rebuttal:
I decided to maintain my score since the current version of this paper is difficult to follow for readers without prior knowledge and thus requires further revision.

**Claims And Evidence:**

The authors could elaborate on the necessity of employing the proposed continuous-time ODE analysis framework to derive algorithms aimed at addressing challenges related to data and computational heterogeneity.

**Essential References Not Discussed:**

This paper have discussed sufficient references.

**Experimental Designs Or Analyses:**

One of the key contributions of the proposed method is addressing the challenge of heterogeneous distributions in FL. While the non-IID data splitting discussed in this paper primarily focuses on label distribution skew, it would be valuable to explore whether the proposed method can also be evaluated under scenarios involving feature distribution skew.

**Methods And Evaluation Criteria:**

Yes, the proposed methods and evaluation criteria make sense for the problem or application at hand.

**Other Comments Or Suggestions:**

The diagrams in Figures 2 and 3 are somewhat roughly illustrated. Specifically, the blue and green curves extend beyond the vertical axis.

**Other Strengths And Weaknesses:**

**Strengths:** This paper presents a novel and motivated theoretical framework based on circuit theory an ODE to analyze the federated optimization process, offering an intriguing and insightful perspective.

**Weaknesses:** The paper heavily relies on circuit theory and continuous-time ODE analysis, which may be less accessible and less reader-friendly for those unfamiliar with these fields.

**Questions For Authors:**

1. In Equation (15), could the authors clarify which client's relative dataset size the notation $p_i$ corresponds to?

2. Does the calculation of solving Equation (33) in the proposed method incurs additional computational costs?

**Relation To Broader Scientific Literature:**

This paper introduces a theoretical framework based on circuit theory and continuous-time ODE analysis, offering the community a novel perspective for understanding the federated optimization process.

**Theoretical Claims:**

1. The paper can be challenging to follow at times due to its reliance on prerequisites in physics and ODE. It would be beneficial if the authors could provide more intuitive explanations of their theoretical framework and devised methods to enhance accessibility for a broader audience.

2. The authors could discuss the essential distinctions in adapting the theoretical framework proposed in (Agarwal and Pileggi, 2023) to the scenario of partial client participation considered in this paper, including the associated technical challenges and how the novel theoretical tools provided in this work addresses them.

---

> ### Author Rebuttal · Authors · 2025-04-01
>
> We would like to thank the reviewer for their comments and hope the following addresses their concerns.
>
> ---
>
> ## **Continuous-time ODE**
>
> The key innovation of FedECADO over the framework proposed by Agarwal and Pileggi (2023) is its introduction of circuit- and simulation-based techniques to address the unique challenges of heterogeneous computation in federated learning. While the previous framework was designed for traditional distributed optimization, assuming homogeneous and always-available worker nodes, FedECADO is specifically designed to handle heterogeneous clients with varying computational capabilities, learning rates, non-IID data distributions, and availability.
>
> At the core of FedECADO is a continuous-time ODE model, which reframes the challenges of heterogeneous client learning rates as a distributed simulation problem. In this model, each client's sub-circuit is defined by local loss function and dataset. By modeling federated learning as a continuous-time ODE, it becomes apparent that client drift due to heterogeneous client computation (where each client has different learnings) is a result of simulating each client for different time-scales.
>
>
> Building on this insight, FedECADO maps the federated learning process to an equivalent circuit to enable the use of circuit simulation to handle the challenges of asynchronous distributed updates. Additionally, this circuit model introduces a new way to model the client sensitivity using a Thevenin impedance, a concept well-established within circuit literature. As part of the novelty of FedECADO, we were able to integrate this sensitivity into the central agent step to improve the convergence rate of the overall federated learning process.
>
>
> The mapping to continuous-time ODEs and circuit models is outside the realm of this community, however, brings new a perspective to address the challenges in federated learning. In this regard, we will add a background section to the supplementary material that covers ordinary differential equations and an understanding of circuit theory to provide the reader with more appreciation on how we derived our methodology.
>
> ---
>
> ## **Dataset size**
>
> The value of $p_i$ refers to the relative number of samples of each client’s training set. The relative weighting allows us to ensure clients with larger datasets have greater influence on the central agent server.
>
>  ---
>
> ## **Computational Cost of (33)**
>
> The main computational cost of performing FedECADO’s central agent step involves computing the Backward-Euler integration step in equation (33). We leverage a constant sensitivity model for each client to ensure that the left-hand side matrix in equation (33) is constant over the simulation. This allows us to pre-LU factor the large matrix prior to training process. During the training process, each epoch of the central agent step performs a forward-backward substitution of the pre-computed LU factor. This helps reduce the computational complexity of FedECADO as well as the overall runtime. The difference in the computational runtime is highlighted in Appendix D.

---

### Official Review · Reviewer_uRW5 · 2025-03-12

**Overall Recommendation:** 3

**Summary:**

This paper considered the federated learning problem, and focused on addresses the challenges from heterogeneous data distributions and computational workloads. To address these challenges, this paper proposed FedECADO, which is the first algorithm that leverages the idea of a dynamical system representation of the federated learning process. The performance of FedECADO is compared with various state-of-the-art methods such as FedProx, FedNova, FedExp, FedDecorr and FedRS.

-----------------update after rebuttal---------------

My concerns were addressed by the authors and I maintain my original score.

**Claims And Evidence:**

Several claims have been made in the paper. Some examples are as follows:

- Claim: The proposed FedECADO uses multi-rate integration to handle heterogeneous client computation
- Evidence: This is validated via experiments. For example, in Table 2, it shows that when clients have varying learning rates and epochs, the accuracy can be improved.

- Claim: The convergence of FL is improved by FedECADO in non-iid settings.
- Evidence: Again, this is validated via experiments. For example, in Table 1, it shows that FedECADO outperforms the considered baselines in different datasets.

**Essential References Not Discussed:**

This paper covers most of the relevant works.

**Experimental Designs Or Analyses:**

The experimental designs are reasonable and similar to many existing works in the federated learning domains.

**Methods And Evaluation Criteria:**

The methods applied in this paper is novel. It should be one of the first work to propose a dynamical system formulation of federated learning.

The evaluation of the proposed algorithm is appropriate as in many existing works in this area, e.g., using accuracy as a performance metric on CIFAR-10/100 datasets, comparisons with state-of-the-art baselines (e.g., FedProx, FedNova, FedExp, FedDecorr and FedRS).

**Other Comments Or Suggestions:**

- Communication cost is another important metric in federated learning settings. What's the advantage of FedECADO over baselines from this perspective?
- Another real-world setting is that the clients/agents in federated learning are dynamic, i.e., some new agents may join the system, while some may leave the system. Can FedECADO handle this setting?

**Other Strengths And Weaknesses:**

Strengths:
- The methods applied in this paper is novel. It should be one of the first work to propose a dynamical system formulation of federated learning.
- FedECADO is proved to be a contraction mapping and ensuring convergence.
- Extensive experimental results were provided to validate the performance of FedECADO

Weakness:
- The design of FedECADO or its performance gain relies on approximating Hessians. This is somehow heuristic. Will this be easily generalized over different tasks/domains/scenarios?
- The ablation study on hyperparameters can be improved, e.g., how to tune the step sizes?

**Questions For Authors:**

- Communication cost is another important metric in federated learning settings. What's the advantage of FedECADO over baselines from this perspective?
- Another real-world setting is that the clients/agents in federated learning are dynamic, i.e., some new agents may join the system, while some may leave the system. Can FedECADO handle this setting?

**Relation To Broader Scientific Literature:**

This paper is related to federated learning, which has been extensively studied over the past decade.

**Theoretical Claims:**

Theorem 4.1 is the main theoretical claim in this work, i.e., it shows that FedECADO is a contraction mapping and hence it ensures convergence.

I checked the proofs in Appendix A, which seems correct.

---

> ### Author Rebuttal · Authors · 2025-04-01
>
> We would like to thank the reviewer for their comments.
>
> ---
>
> ## **Hessian Approximation**
>
> In federated learning with non-IID data distribution, each client’s local loss function is a function of its unique dataset. The Hessian captures the curvature of each loss function at the client’s specific operating point. As a result, different tasks, domains and scenarios will create unique solution landscapes whose curvature can be measured using the Hessian. We use this Hessian to measure how sensitive a client’s model is to changes from the global model during the central aggregation step.  This sensitivity helps the server anticipate the impact of model aggregation on individual clients.
>
>
> We approximate a constant Hessian for each client by sampling multiple data points and averaging their respective Hessians.  Future work will improve this approach by using prior work (such as [R2]-[R4]) to improve efficiency and accuracy.
>
>
>  ---
>
> ## **Ablation Study**
>
> In this work, the server step-sizes are modeled as time-steps for a Backward-Euler integration. These step sizes are adaptively selected based on the local truncation error, a measure of how closely we are tracking the designed ordinary differential equation. As a result, we do not select the step sizes, but rather select the tolerance for the discretization error due to Backward-Euler integration, denoted as $\delta$ in equation (36). We find that the value of $\delta$ does not provide much difference in the final simulation accuracy. We have added a figure, shown in [R1] where $\delta$ is changed by three orders of magnitude (assuming a fixed client step size) for training on CIFAR-10 dataset. From the figure, we notice that the convergence plots remain well-behaved regardless of the choice of $\delta$. Future work will look at designing the value of $\delta$ based on approximating the Lipchitz constant of the gradient-flow updates.
>
> ---
>
> ## **Communication Cost**
>
> The advantage of FedECADO in terms of communication is that we can handle client updates with heterogeneous client learning rates with only an additional scalar constant being communicated at each epoch. This scalar constant, $\Delta T$, measures the amount of time each client is simulated for.
>
>
> ---
>
> ## **Dynamic Clients**
>
> Adding and removing clients during the federated learning process can definitely be mapped to FedECADO's circuit simulation framework. We can model this behavior similarly to the switching of transistors in circuits, where transistors are added in series with clients and are “turned on or off” to connect or disconnect clients from the central agent. Circuit simulators, with their deep history in simulating discrete events, are well-suited for this type of modeling.
>
>  ---
>
> [R1] https://drive.google.com/file/d/1OZRDtok0Ou6RdELJ2Eb13N7_4TJaU-J3/view?usp=share_link
>
> [R2] Elsayed, M., Farrahi, H., Dangel, F. and Mahmood, A.R., 2024. Revisiting scalable hessian diagonal approximations for applications in reinforcement learning. arXiv preprint arXiv:2406.03276.
>
> [R3] Elsayed, M. and Mahmood, A.R., 2022. Hesscale: Scalable computation of hessian diagonals. arXiv preprint arXiv:2210.11639.
>
> [R4] Yao, Z., Gholami, A., Keutzer, K. and Mahoney, M.W., 2020, December. Pyhessian: Neural networks through the lens of the hessian. In 2020 IEEE international conference on big data (Big data) (pp. 581-590). IEEE

---

### Official Review · Reviewer_xyRN · 2025-03-18

**Overall Recommendation:** 3

**Summary:**

This paper proposes a federated variant of ECADO (Agarwal and Pileggi, 2023). ECADO is an equivalent circuit approach to distributed optimization. ECADO consists in  reconstructing a distributed optimization problem in terms of circuit principles, and finding the critical points of the equivalent circuit model using a distributed Gauss-Seidel (G-S) process.

The main contribution of this paper is the multi-rate numerical integration for heterogeneous computation, inspired by asynchronous distributed circuit simulation (White and Sangiovanni-Vincentelli, 2012). This is meant to address asynchronous local updates in federated learning resulting from from clients with heterogeneous computational capabilities.

The paper provides numerical simulation illustrating the performance of the proposed FedECADO approach.

**Claims And Evidence:**

The claim that FedECADO outperforms other federated optimization techniques is not well-supported. For example, FedRS, FedExp, and FedDecorr significantly outperform FedECADO on  CIFAR-10 dataset (Table 1). Moreover, FedECADO does not significantly outperform FedRS in Table 2.

The paper does not show the convergence curves for all compared methods.

**Essential References Not Discussed:**

N/A

**Experimental Designs Or Analyses:**

See "Methods And Evaluation Criteria"

**Methods And Evaluation Criteria:**

Evaluation settings are standard in the context of federated learning. One could argue that more datasets/models are needed, but I think that the paper has already a sufficient number of datasets/models is enough.

I argue that in this optimization-focused paper, providing convergence curves, in addition to final accuracy, is beneficial. Moreover, it would be helpful to illustrate the performance of the proposed method on a synthetic dataset, where the characteristics of the optimization problem are adjustable.

**Other Comments Or Suggestions:**

I think it would be better to use $I_{i}^{k}$, instead of $I_{L}^{i^k}$.

**Other Strengths And Weaknesses:**

My main concern about this paper is its relative lack in terms of novelty. The paper is in many parts mirroring (Agarwal and Pileggi, 2023). The contribution of the paper is in Section 4.2. This contribution is an heuristic heavily inspired by (White and Sangiovanni-Vincentelli, 2012).

I find that the paper is not well-written, and it was not easy to read through and to follow it without reading (Agarwal and Pileggi, 2023). The paper did not do a good job in summarizing and explaining (Agarwal and Pileggi, 2023), which is expected given that (Agarwal and Pileggi, 2023) is not a popular paper, and the reader is not supposed to know it a-priori.

FedECADO has a 6% computational increase over FedRS (Table 6).

**Questions For Authors:**

N/A

**Relation To Broader Scientific Literature:**

The paper discusses relevant federated optimization papers.

**Theoretical Claims:**

The only real theoretical claim of the paper is Theorem 4.1. I did not check the correctness of the proof, but the result sounds intuitively correct.

---

> ### Author Rebuttal · Authors · 2025-04-01
>
> Thank you for your comments.
>
>  ---
> ## **Novelty of FedECADO**
>
> Regarding novelty, FedECADO is inspired by distributed circuit simulation, where sub-circuits are independently simulated and recombined using waveform relaxation (White and Sangiovanni-Vincentelli, 2012). Our approach extends waveform relaxation by incorporating inductors and constant sensitivity models to accelerate convergence. The key innovation lies in establishing a connection between federated learning and circuit simulation, allowing simulation principles to be adopted to federated learning.
>
>
> FedECADO extends the circuit model of distributed optimization presented in (Agarwal and Pileggi, 2023) by introducing multi-rate integration and chord-based modeling to address the unique challenges of federated learning. Unlike in distributed optimization, where the worker nodes are homogeneous and always available, federated learning can have heterogeneous client computation capabilities where computational nodes are not available simultaneously and exhibit varying local learning rates. The multi-rate integration method is a key innovation of our work that can handle these new challenges. Importantly, the circuit model provides a new perspective to these challenges (such as framing heterogeneous client learning rates as asynchronous communication), which enables the development of intuitive and effective heuristics. FedECADO provides the first bridge between these fields, enabling new research directions where physical principles and simulation methods can drive federated learning solutions.
>
>  ---
> ## **Optimization Plots and Update on Table 1**
>
> To the reviewer’s request, we have added the plots for training CIFAR-10 in the following link (in addition to the supplementary):
>   https://drive.google.com/file/d/1Ye1AiKFR2c8Xz2Tmps6x387WG8l0xlte/view?usp=share_link
>
> We also would like to address the inconsistency between the text and Table 1. We have an unfortunate typo in the previous version and have have updated Table 1 (specifically FedRS, FedDecorr, and FedExp) to match the given plots.
>
>
> | Classification Acc. (%) | FedECADO   | FedNova    | FedProx    | FedExp     | FedDecorr  | FedRS      |
> |-------------------------|------------|------------|------------|------------|------------|------------|
> | Mean (Std.)             | 57.8 (3.6) | 48.9 (2.9) | 44.3 (3.2) | 45.3 (4.7) | 45.3 (4.7) | 45.3 (4.7) |
>
>
>  The convergence plots for this CIFAR-10 example is provided in the link below:
>
>
> https://docs.google.com/document/d/e/2PACX-1vTamA9cixTmdcZecNeVjy7PZ9i-NuQi9c53d0zG2wPzjTg2tuoF4K_aGFIVZgLt096D0189JNK235ht/pub
>
> ---
> ## **Background on circuit details**
>
> We understand that the inspiration for FedECADO (namely circuit-based ODEs) is not within the realm of the community. We plan on adding additional supplementary information that gives a primer on circuit physics and simulation in the appendix.

---

### Official Review · Reviewer_jW9j · 2025-03-23

**Overall Recommendation:** 3

**Summary:**

This paper explores the interpretation of federated learning in dynamical systems and adapts the ECADO algorithm to federated learning, proposing the FedECADO algorithm. It uses a physical equivalent circuit model to explain the federated learning process and targets optimizations for non-IID data and client asynchronous updates.

**Claims And Evidence:**

The proposed FedECADO method aims to solve the classic issues of non-IID data and client asynchronous updates in federated learning.

**Essential References Not Discussed:**

None

**Experimental Designs Or Analyses:**

Around line 366, this paper claims that Table 1 demonstrates that FedECADO has the highest average accuracy. However, this claim clearly contradicts the actual results shown in Table 1.

**Methods And Evaluation Criteria:**

This paper uses the Aggregate Sensitivity Model to address the non-IID data distribution issue in federated learning. However, as mentioned in lines 32 and 233 of the paper, the Aggregate Sensitivity Model only reflects differences in the number of client datasets and does not address optimization for data distribution differences.

Furthermore, the Multi-rate integration with adaptive step sizes, used to address asynchronous client updates, has limited applicability, especially for scenarios where model updates are not done locally but are aggregated on the server, with clients only accepting the global model parameters provided by the server.

Additionally, the core idea of the method is derived from ECADO, which limits its novelty.

**Other Comments Or Suggestions:**

• The formatting on the first page is incorrect.

• The layout of Table 4 and Table 6, as well as most of the formulas, can be improved.

**Other Strengths And Weaknesses:**

As noted earlier, the overall clarity of the paper could be improved.

**Questions For Authors:**

How does the Aggregate Sensitivity Model module address the issue of non-IID data distributions in federated learning?

**Relation To Broader Scientific Literature:**

Using equivalent circuits to explain the federated learning is a novel perspective.

**Theoretical Claims:**

I am not familiar with EC-related work, so I cannot determine if the process of using EC to model federated learning in this paper is correct, but it generally makes sense.

---

> ### Author Rebuttal · Authors · 2025-04-01
>
> We would like to thank the reviewer for their comments.
>
> ---
>
> ## **Novelty of FedECADO**
>
>  FedECADO builds on the distributed optimization method,  ECADO, and introduces key innovations (multi-rate integration and chord-based modeling) to address the unique challenges in federated learning. Unlike in distributed optimization, where the worker nodes are homogeneous and always available, we tackle heterogeneous client computation capabilities where computational nodes are not available simultaneously and exhibit varying local learning rates and non-IID datasets. We believe the novelty of our approach lies in establishing a connection between federated learning and circuit methods, which offers a new perspective on these challenges.
> For example, we re-interpret heterogeneous client learning rates as sub-circuits simulating for different time-periods. This allows us to take inspiration from circuit methods for distributed simulation. While advancements in general optimization have made the connection with circuits to inspire fast-convergent methods [R1]-[R3], FedECADO is the first work that introduces circuit knowledge to federated learning. This provides insights that are easily attained from the circuit model and simulation practices.
>
> ---
>
> ## **Aggregate Sensitivity Model**
>
> Could the reviewers clarify their specific concerns about aggregate model sensitivity?
>
> We would like to emphasize that in federated learning with non-IID data, each client’s local loss function has a unique solution landscape shaped by its data distribution.  The Hessian captures the curvature of the loss function at the client’s specific operating point and local training data. We use the Hessian to estimate the first-order sensitivity of a client’s local model to changes in the global model parameters. This sensitivity is used to anticipate how aggregation steps will affect individual client states. To approximate a representative Hessian for clients, we sample multiple data points and compute an averaged Hessian. Future work in FedECADO can also extend this by approximating the Hessian via [R4]-[R6]
>
> ---
>
> ## **Multi-Rate Integration**
> Could the reviewers clarify their concern regarding multi-rate integration?
>
> To clarify, we are not considering asynchronous client updates where clients begin with different global model versions. Instead, using the circuit model, we interpreted the challenge of heterogeneous client learning rates as clients simulating their local models for different time scales. To remedy this, the central server performs multi-rate integration by accepting the latest client updates, similar to FedAvg. Then, rather than directly aggregating these updates as in FedAvg, FedECADO first applies a linear interpolation/extrapolation operator to synchronize the simulation time scales across clients. The central agent step is then computed using a Backward-Euler integration step. This approach has a process similar to the aggregation step in FedAvg, but adds the additional linear operation, $\Gamma(\cdot)$, to address the challenge of heterogeneous client computation, and uses a Backward-Euler step to maintain numerical stability.
>
> Additionally, in FedECADO, each client also receives the updated global state as shown in Algorithm 2 in the main text.
>
> ---
>
> ## **Improving the background on circuit-based framework**
>
> As part of improving the readability, we will include a section in the appendix that provides a deeper background on ODEs and circuit analysis.
>
> ---
>
> ## **Formatting**
>
> Thank you for the suggestion. In the following revision, we will improve the layout of Tables 4 and 6 as well as the equations.
>
> ---
>
> ## **Update on Table 1 and Optimization Plots**
> Thank you for pointing out the inconsistency in Table 1. This was an unfortunate type in the previous version and is updated to reflect true classification accuracy as shown in the response for Reviewer xyRN.
>
> ---
> [R1] Boyd, S., Parshakova, T., Ryu, E. and Suh, J.J., 2024. Optimization algorithm design via electric circuits. Advances in Neural Information Processing Systems, 37, pp.68013-68081.
>
> [R2] Agarwal, A., Fiscko, C., Kar, S., Pileggi, L. and Sinopoli, B., 2022. ECCO: Equivalent Circuit Controlled Optimization. arXiv preprint arXiv:2211.08478.
>
> [R3] Yu, Y. and Açıkmeşe, B., 2020. RC circuits based distributed conditional gradient method. arXiv preprint arXiv:2003.06949.
>
> [R4] Elsayed, M., Farrahi, H., Dangel, F. and Mahmood, A.R., 2024. Revisiting scalable hessian diagonal approximations for applications in reinforcement learning. arXiv preprint arXiv:2406.03276.
>
> [R5] Elsayed, M. and Mahmood, A.R., 2022. Hesscale: Scalable computation of hessian diagonals. arXiv preprint arXiv:2210.11639.
>
> [R6] Yao, Z., Gholami, A., Keutzer, K. and Mahoney, M.W., 2020, December. Pyhessian: Neural networks through the lens of the hessian. In 2020 IEEE international conference on big data (Big data) (pp. 581-590). IEEE

---

### Decision · Program_Chairs · 2025-05-01

**Decision:**

Accept (poster)

**Comment:**

All reviewers tend to accept this paper (scores 3, 3, 3, 3) and reach a consensus on the following aspects:
1. This paper provides a new approach to federated learning built on ECADO (Agarwal and Pileggi, 2023), which is an equivalent circuit approach to distributed optimization.
2. This paper is difficult to follow as it lacks sufficient self-contained understandable background knowledge of (Agarwal and Pileggi, 2023).

Reviewer jW9j and Reviewer xyRN have concerns on the novelty of FedECADO, worrying that FedECADO may be a straightforward application of ECADO (Agarwal and Pileggi, 2023) to the FL setting. The authors clarified in the rebuttal:
> Unlike in distributed optimization, where the worker nodes are homogeneous and always available, federated learning can have heterogeneous client computation capabilities where computational nodes are not available simultaneously and exhibit varying local learning rates.

Despite the new perspective to applying physical simulation principles to solve federated learning problems, all reviewers found it extremely challenging to follow as it heavily relies on prerequisites in physics and ODE. I’m inclined to recommend acceptance on the condition that the authors will provide more intuitive explanations of their theoretical framework and make the paper self-contained.